# Estimating the risk and spatial spread of measles in populations with high MMR uptake: Using school-household networks to understand the 2013 to 2014 outbreak in the Netherlands

**James D. Munday** [1,2,3] *, **Katherine E. Atkins** [1,2,4], **Don Klinkenberg** [5], **Marc Meurs** [6], **Erik Fleur** [6], **Susan JM Hahné** [5], **Jacco Wallinga** [5,7], **Albert Jan van Hoek** [1,2,5]

**1** Centre for Mathematical Modelling of Infectious Diseases, London School of Hygiene and Tropical Medicine, London, United Kingdom, **2** Department of Infectious Disease Epidemiology, London School of Hygiene and Tropical Medicine, London, United Kingdom, **3** Department of Biosystems Science and Engineering, ETH Zürich, Basel, Switzerland, **4** Usher Institute, College of Medicine and Veterinary Medicine, University of Edinburgh, Edinburgh, United Kingdom, **5** National Institute for Public Health and the Environment (RIVM), Bilthoven, the Netherlands, **6** Education Executive Agency (DUO), The Hague, the Netherlands, **7** Department of Biomedical Data Sciences, Leiden University Medical Centre, Leiden, the Netherlands

* james.munday@bsse.ethz.ch

**Data Availability Statement:** The code and school network data is available at https://github.com/jdmunday/SchoolsMeaslesNL. School population statistics are publicly available from: https://www.duo.nl/open_onderwijsdata/index.jsp. Measles

## Abstract

### Background

Measles outbreaks are still routine, even in countries where vaccination coverage exceeds the guideline of 95%. Therefore, achieving ambitions for measles eradication will require understanding of how unvaccinated children interact with others who are unvaccinated. It is well established that schools and homes are key settings for both clustering of unvaccinated children and for transmission of infection. In this study, we evaluate the potential for contacts between unvaccinated children in these contexts to facilitate measles outbreaks with a focus on the Netherlands, where large outbreaks have been observed periodically since the introduction of mumps, measles and rubella (MMR).

### Methods and findings

We created a network of all primary and secondary schools in the Netherlands based on the total number of household pairs between each school. A household pair are siblings from the same household who attend a different school. We parameterised the network with individual level administrative school and household data provided by the Dutch Ministry for Education and estimates of school level uptake of the MMR vaccine. We analysed the network to establish the relative strength of contact between schools and found that schools associated with low vaccine uptake are highly connected, aided by a differentiated school system in the Netherlands (Coleman homophily index (CHI) = 0.63). We simulated measles outbreaks on the network and evaluated the model against empirical measles data per

case data is aggregated from clinical records and owned by the Dutch National Institute for Public Health and the Environment (RIVM) and is not publicly available. Requests for access can be made directly to RIVM by emailing osiris.aiz@rivm. nl.

**Funding:** This study is funded by the National Institute for Health and Care Research (NIHR) Health Protection Research Unit in Vaccines and Immunisation (NIHR200929), a partnership between UK Health Security Agency and the London School of Hygiene and Tropical Medicine and University of Cambridge. JM, AJvH and KEA received funding from the National Institute for Health Research (NIHR, https://www.nihr.ac.uk/) Health Protection Research Unit in Vaccines and Immunisation (NIHR200929). The remaining authors received no specific funding for this research. The funder did not play any role in the study design, data collection and analysis, decision to publish, or preparation of the manuscript.

**Competing interests:** The authors have declared that no competing interests exist.

**Abbreviations:** CHI, Coleman Homophily Index; CI, Credible interval; HI, Homophily Index; IQR, Interquartile Range; MMR, mumps, measles and rubella; MPP, mean pairwise probability; ROC, receiver operating characteristic; wROC, weighted receiver operating characteristic.

postcode area from a large outbreak in 2013 (2,766 cases). We found that the network-based model could reproduce the observed size and spatial distribution of the historic outbreak much more clearly than the alternative models, with a case weighted receiver operating characteristic (ROC) sensitivity of 0.94, compared to 0.17 and 0.26 for models that do not account for specific network structure or school-level vaccine uptake, respectively. The key limitation of our framework is that it neglects transmission routes outside of school and household contexts.

## Conclusions

Our framework indicates that clustering of unvaccinated children in primary schools connected by unvaccinated children in related secondary schools lead to large, connected clusters of unvaccinated children. Using our approach, we could explain historical outbreaks on a spatial level. Our framework could be further developed to aid future outbreak response.

## Author summary

- Why was this study done?

  ○ The measles, mumps and rubella (MMR) vaccine is very effective at protecting against measles infection; however, outbreaks are still possible especially when larger numbers of unvaccinated people group together allowing longer transmission chains to form.

  ○ In the Netherlands, 2 groups in particular are associated with low vaccine uptake, and these groups are associated with particular schools in the differentiated school system, creating clusters of unvaccinated children in these schools.

  ○ We wanted to understand if these schools are also well connected to each other through relationships in shared households (e.g., siblings that attend 2 separate schools) and what this means for outbreak risk in the Netherlands.

- What did the researchers do and find?

  ○ We constructed a network of all the schools in the Netherlands and quantified how connected each pair of schools is by counting the number of children in one school that share a household with a child in the other school.

  ○ We used this data to assess how connected the schools that were associated with low vaccine uptake were to each other and constructed a model of disease transmission between schools to assess the implications of this connectivity.

  ○ We found that these schools were more connected to each other than would be expected based on the patterns of connectivity in the rest of the network (Coleman homophily index (CHI) of 0.63) and that this additional connectedness was important in explaining the nature of a past measles epidemic.

- What do these findings mean?

  ○ Our findings highlight the role of the structure of the differentiated school system in allowing large outbreaks to occur and offer an explanation for why outbreaks in the

Netherlands tend to be so large compared to other countries with similar vaccine uptake.

○ The framework we use can be adapted to evaluate ongoing risk of measles outbreaks by re-parameterising with updated school-connectivity and renewed estimates of susceptibility in each school by combining historical outbreak and vaccine uptake data.

○ The main limitation of our approach is that it only considers schools and household contexts and does not account for infectious transmission outside of these settings.

## Introduction

The World Health Organization has outlined ambitious goals for eliminating measles [1]. Despite the effectiveness of the measles, mumps, and rubella (MMR) vaccine, estimates suggest that about 95% of the population needs to be vaccinated to achieve herd immunity, which is a level that prevents widespread disease transmission [2–6]. However, many countries are facing challenges due to increasing numbers of unvaccinated children. This rise is partly driven by concerns of parents and guardians regarding routine childhood vaccines, in particular, MMR vaccination rates declined during and after the COVID-19 pandemic [7,8]. This increase in unprotected children increases the risk of measles outbreaks as has been documented in the past for measles and other highly infectious pathogens [9–12].

Notably, even in countries where more than 95% of people have received at least 1 dose of MMR, measles outbreaks still occur. This is because the traditional concept of herd immunity assumes that vaccines are distributed evenly throughout the population [13]. In reality, the tendency to decide not to vaccinate can cluster within specific social groups [14], underscoring the need to understand how unvaccinated children interact with each other [15]. Outbreaks pose immediate risks to vulnerable populations, including young children who cannot yet receive vaccinations and people with certain medical conditions.

The phenomenon of clustering of unvaccinated individuals within social groups in the Netherlands is well documented [16], where the overall uptake has been high (>98% MMR1 by age 2) for nearly 30 years [17]. Among the unvaccinated population, there are similar associations as other high-income settings, e.g., lower income and migration status. In addition to these, however, there are 2 subgroups who have a stronger association with low vaccination coverage. These are members of the Orthodox Protestant ("*Reformatorisch*") community, a denomination of the Christian faith and among those who associate with Anthroposophic ("*Antroposofisch*") views and philosophies [18,19]. Both of these groups form social clusters to some degree, particularly within schools. The Netherlands has a differentiated school system, which allows parents to organise and/or choose primary and secondary schools freely, often in line with their household values or faith. Approximately a third of state-funded primary and secondary schools are organised by the government and do not promote any specific religious or ideological stance (the identity referred to as "Openbaar"). The remaining two thirds of state-funded schools are affiliated with one of 27 distinct identities linked to a faith or (educational) philosophy. An analysis of vaccine coverage in Dutch schools in 2013 showed that schools with an Anthroposophic and Orthodox Protestant identity had a higher concentration of unvaccinated children who had not yet received at least 1 dose of the MMR vaccine [20].

Since the introduction of MMR vaccine in 1989, the Netherlands has observed various measles outbreaks. From smaller outbreaks, like in 2008, 2011, 2018, and 2019 [21] to larger

outbreaks occurring in 1999 to 2000 and 2013 to 2014, which each totalled an infection count of the order of 30,000, with cases distributed across the country [22]. In both large outbreaks, the majority of infections reported were in children, with 77% of cases in 2013/14 among 4 to 17 year olds [17]. Interestingly, variation in outbreak sizes [17,23–26] suggest that factors beyond clustering of unvaccinated children within individual schools play a role. Concretely, the clustering in individual schools can explain the smaller locally concentrated outbreaks like the outbreak in Anthroposophic schools in 2008 [24], but is not sufficient to explain the very large and geographically disparate outbreaks witnessed in 1999 and 2013 [17,26]. We therefore suggest that focusing solely on vaccine uptake within schools overlooks the potential impact of interactions among unvaccinated siblings. Families may align their choice of primary and secondary schools with a particular identity, which has the potential to lead to larger clusters of susceptible individuals that cover broader geographical areas than individual school boundaries. To understand this clustering better and assess its implications for larger outbreaks, we propose a method that examines connections between schools and households using national school registration data. This approach builds on previous research showing that households and schools are primary locations for close and lasting contacts, which play a significant role in disease transmission [27,28]. We have previously shown the effectiveness of using school-household data to study how the structure of the education system relates to the spread of infectious diseases among school-age children [29]. This approach's reliability has also been confirmed through analysing data related to COVID-19 in the Netherlands [30].

Similarly to Munday and colleagues [29], we constructed a comprehensive network that links schools and households with unvaccinated individuals. For this we have used data from the Dutch Ministry of Education to establish household connections between schools. We first analysed the network to establish the connectedness of schools by identity quantifying homophily indices, comparing networks with geographical distances between schools and by evaluating the nature of clusters identified using community detection. We then extended our approach to include vaccination informed by vaccine uptake estimates at the school level from Klinkenberg and colleagues [20]. Combining these data, we aimed to better understand how unvaccinated children interact. To validate the usefulness of this network for assessing outbreak risks and their potential impact, we simulated measles outbreaks within this network and compared the results to the large outbreak observed in 2013/14 [22]. To assess the robustness of this combined network, we compare our results with 2 alternative combined network structures, the first in which MMR uptake is not clustered by school but by postcode, and the second in which schools are not connected based on school-households contacts but based on geographical distance.

## Methods

### Ethics statement

Ethical approval for this work was obtained from the London School of Hygiene and Tropical Medicine (16028–1).

### School network

Children in the Netherlands enter primary school at the age of 4, where they remain for 8 years. At the age of 12 they enter secondary school, which is usually separate from primary. Secondary school is divided into 3 academic tiers: VMBO (Pre-vocational training), HAVO (General senior education), and VWO (Pre-university). Students remain studying in these contexts until they are 16, 17, and 18 years old, respectively.

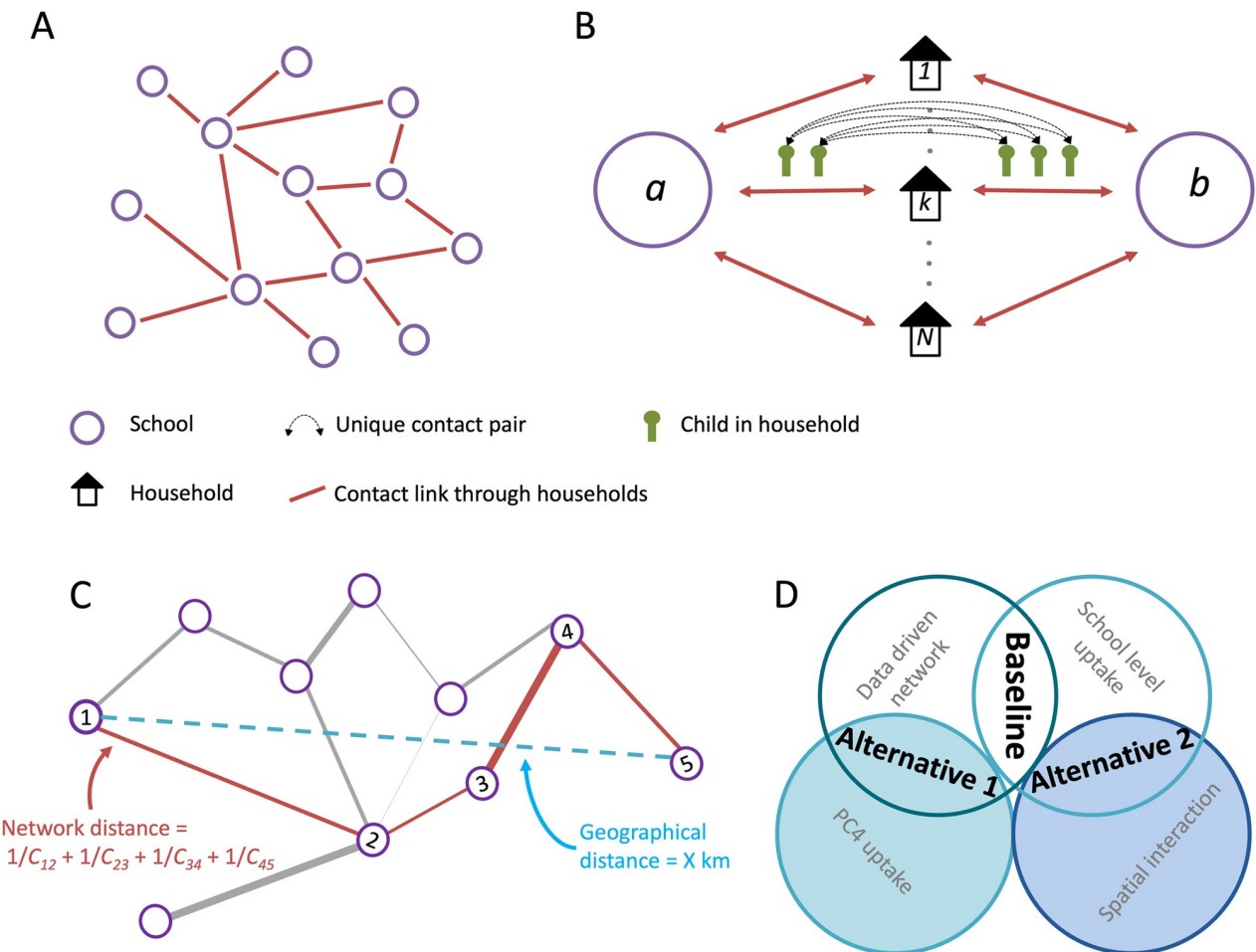

**Fig 1.** (A) Network of schools constructed such that schools are connected when contact is made between pupils of different schools within a household. (B) The strength of contact between schools is quantified by calculating the number of unique contact pairs (1 child in each school). The number of pairs per household is the product of the number of children who attend school $a$ and the number of children who attend school $b$. The total number of unique pairs is the sum of unique pairs in each household with children attending both school $a$ and $b$. Figure adapted from [29]. (C) Calculation of network distance between nodes 1 and 5 is the sum of the edges along the shortest path between those nodes. (D) Schematic of the components of the different network models. The baseline used the data driven contact network and school level uptake. Alternative models: 1. Uptake based on four-digit postcode areas of the children who attend the school. 2. Interaction based on a spatial interaction kernel.

To construct a network of schools as connected through household contacts (Fig 1A), we followed the approach we described in Munday and colleagues [29]. We used government data to calculate the total number of unique contact pairs between schools for the reference date of 1 October 2013 (the year of the last major measles outbreak in the Netherlands). The Ministry of Education (Dienst Uitvoering Onderwijs, DUO) holds data on school attendance for each child in non-private education for the Netherlands (>99% of school-aged children). From these data, the ministry calculated the number of children per individual school at each unique address (irrespective of class, age, and gender). For each unique address, based on the number of children per school, the number of unique contact pairs between each pair of schools represented in that address was calculated, where a contact pair is a pair of students who live in the same address but attend different schools (for example, 2 children in school A and 2 children in school B form $2 \times 2 = 4$ unique contact pairs between school A and B) (Fig 1B). Subsequently, for each pair of schools in the data, we calculated the total number of

contacts across all addresses, resulting in a total number of unique contact pairs between each pair of schools (for example, 46 contact pairs between school A and B). In this calculation, schools are defined as school-locations identified by their location number in the DUO database. Therefore, we distinguish between the multiple (geographical) school buildings/locations even though they belong to the same school administration. The resulting network describes the number of unique contact pairs between each pair of schools in the Netherlands, including between primary and secondary schools.

## School data

The school identity (see **Table A in S1 Text**), the total number of students per school (also from October 2013) and the catchment of each school, the number of students per four-digit postcode (PC4) was obtained from open source education data [31]—Dutch postcodes are an alphanumeric code of 4 digits followed by 2 letters. The exact geographical location of each school was obtained from the full address and postcode of the school site.

## Community structure in the school network

Firstly, to evaluate the connectedness of schools irrespective of the identity, we evaluated the community structure within the school network. Communities represent groups of schools that are more connected to each other than to other schools on the network. For this, we used the modularity maximisation-based Leiden algorithm, implemented in the *leidenalg* python package [32]. This approach partitions the network such that groups of schools are identified, which are more connected to each other than schools that belong to other groups. The size of the groups detected by the Leiden algorithm can be adjusted using the resolution parameter— the higher the value of resolution parameter, the smaller the communities detected (**S1 Text**). We first evaluated partitions made using different values of the resolution parameter. To establish the most meaningful scale of communities, we partitioned the network with values of resolution between 0.1 and 1. We evaluated the partitions against 4 metrics: *Internal edge density* [33], *Modularity density* [34], *Neman Girvan modularity* [35], and *Surprise* [36], for details see **S1 Text**. We established that a resolution parameter of 1 gave the best score in all metrics and therefore proceeded with this for the further analysis. To establish a consensus partition, we generated 20 partitions of the network. From the partitions we calculated a similarity matrix, where each element was equal to the frequency with which each pair of schools was partitioned into the same community. We repeated the process again but instead partitioned the network described by the similarity matrix. We repeated this process until all 20 partitions were identical, we present this partition as the consensus partition. We evaluated each stage of the process by calculating the normalised mutual information of each pair of partitions. We then evaluated the composition of each of the communities in the consensus partition.

To evaluate patterns by geographical location and affiliated school identity in the initially generated partitions, we calculated the mean pairwise probability (MPP) that any 2 schools of the same particular identity or province fall into the same community over the partitioned networks, giving the propensity for schools of a particular identity to form communities in the network.

## Network analysis

We explicitly evaluated the relative connectedness of schools with other schools of the same identity on the network by calculating the Basic Homophily ($H_i$) and Coleman Homophily Index ($IH_i$) (CHI) [37] (Eqs 1 and 2) of each school identity. This measure gives the proportion of neighbouring schools that are of the same identity relative to the prevalence of that identity

in the school system.

$$H_i = \frac{s_i}{s_i + d_i} \tag{1}$$

$$IH_i = \frac{H_i - w_i}{1 - w_i} \tag{2}$$

Where, $s_i$ is the number of connections between schools part of identity $i$, $d_i$ is the total number of connections between schools in identity $i$ and any other school in the network, and $w_i$ is the proportion of schools in the network that belong to identity $i$.

To explore longer-range connections in the network, we compared geographical and network distance between pairs of schools in the network, similarly to previous work by Donker and colleagues [38] studying hospital referral networks. Network distance was defined as the length of the shortest path between schools on the reciprocal contact network. The weights of the edges in the reciprocal network are equal to the reciprocal of the number of unique contact pairs between schools. The network distance between 2 schools is therefore the lowest possible sum of edges that form a path between the schools on the reciprocal network (Fig 1C); i.e., $\sum_{i=1}^{N_{path}-1} \frac{1}{C_{i+1,i}}$, where $C_{i+1,i}$ is the number of contact pairs between consecutive schools $i$ and $i+1$ in a shortest path of $N_{path}$ edges.

We calculated the network distance (distance ratio) and geographic distance (km) of 1,000 randomly sampled pairs of schools from the biggest faith-based school identities in the Netherlands: Roman Catholic ("*Rooms-katholiek*") and Protestant ("*Protestants-Christelijk*"). We also calculated the distances for Orthodox Protestant ("*Reformatorisch*") and Anthroposophic ("*Antroposofisch*") identities for comparison.

Schools with a low distance ratio are more closely connected on the network relative to their geographic distance than schools with higher distance ratios. We calculated distance ratios for pairs of schools of the same identity to that of schools in the rest of the network, defined as all schools not associated with that identity. To account for geographic location of schools, we compared distance ratios for schools sampled from the "rest of the network" from the same two-digit postcode area as each school sampled from the denomination of interest.

## Transmission model

In addition to our analysis of the network of schools, we evaluated the epidemiological relevance of any increased connectedness between schools, specifically concerning outbreaks of measles. We used the network to simulate outbreaks of measles in school-aged children (4 to 17 year olds) (S1 Text). We extended the method used in Munday and colleagues [29] to include vaccination. In a generation-based model, each school could be in one of 3 epidemiological states: susceptible, infected, or recovered. Schools in the susceptible state had immunity among its pupils equal to the complement of the estimated vaccination coverage ($V_j$) in that school in October 2013 estimated by Klinkenberg and colleagues [20] (**Fig G in S1 Text**). When a school becomes infected, the total outbreak size within this school is obtained by a final size equation [39] (Eq 3) considering the total number of students, the number of susceptibles and $R_0$ modified by the proportion susceptible ($1-V_j$) to give the effective reproduction number in the population.

$$R_j(\infty) = (1 - V_j)(1 - e^{-(1-V_j)R_0 R_j(\infty)}) \tag{3}$$

Subsequently, over the course of the school outbreak in school $j$, a new outbreak can be initiated in a connected school $i$, with probability $P_{trans,ij}$ depending on the number of contact

pairs ($C_{ij}$), the probability that a contact from the infected is infected ($P_j^I$) (based on the final outbreak size), the percentage susceptible in the connected school, $P_i^S$, and the probability that introducing an infected child into the susceptible school would lead to an outbreak ($P_i^{OB}$) (see **S1 Text** for full description).

$$P_{trans,ij} = 1 - (1 - P_j^I P_i^S q P_i^{OB})^{C_{ij}} \qquad (4)$$

For our application, we have assumed a Poisson distributed offspring distribution for within-school transmission, hence $P_i^{OB}$ is equal to $P_i^I$ [40]. Outbreaks were simulated from an initial state, where one or more schools were in the infected state and the remaining schools were in the susceptible state. We set the "within school" reproduction number ($R_0$) to 15, consistent with estimates for measles (12–18) [41] and the probability of transmission to a susceptible sibling ($q$) of 0.9 consistent with estimates of a household secondary attack rate of ca. 90% [42]. To assess the sensitivity of the model to these parameters, we repeated the analysis for $R_0$ values of 12 and 18 and for $q$ of 0.5 (Table 1).

## Alternative network models

To evaluate the importance of the specific connectedness in the school network and vaccination at school level to the simulated outbreaks, we designed alternative models which contained only part of the information used in the full model (Fig 1D).

"Alternative model 1" was designed to establish the importance of clustering of unvaccinated children in particular schools. We analysed an alternative parameterization assuming children had a probability of being vaccinated equal to the vaccine uptake of the PC4 where they lived. We used data on the residence of children in each school to calculate the proportion of children in each school who live in each PC4. School vaccination uptake was set as the weighted average of vaccination uptake at PC4 level, weighted by the proportion of children who live in each PC4.

We included "alternative model 2" to assess the importance of the specific network structure defined by the school data to the overall dynamics of outbreaks. We constructed an alternative school contact network where the geographic distance between connected schools followed a similar relationship to the baseline model; however, contact is spread evenly over all schools according to that relationship. The spatial distribution of a school's immediate neighbours in the data-based school network can be described by an exponential distribution. We therefore used this relationship and weighted by the degree of the connecting schools to describe the spatial relationship for alternative model 2. To calibrate the spatial model, we matched the distribution of distance between schools connected by contact pairs to the school data-derived network (details in **S1 Text**).

**Table 1. Transmission model parameters.**

| Parameter | Symbol | Value | Sensitivity study* |
|---|---|---|---|
| Within school reproduction number | $R_0$ | 15 [41] | 12, 18 |
| Probability of transmission between siblings | $q$ | 0.9 [42] | 0.5 |

\* Reported in **S1 Text**.

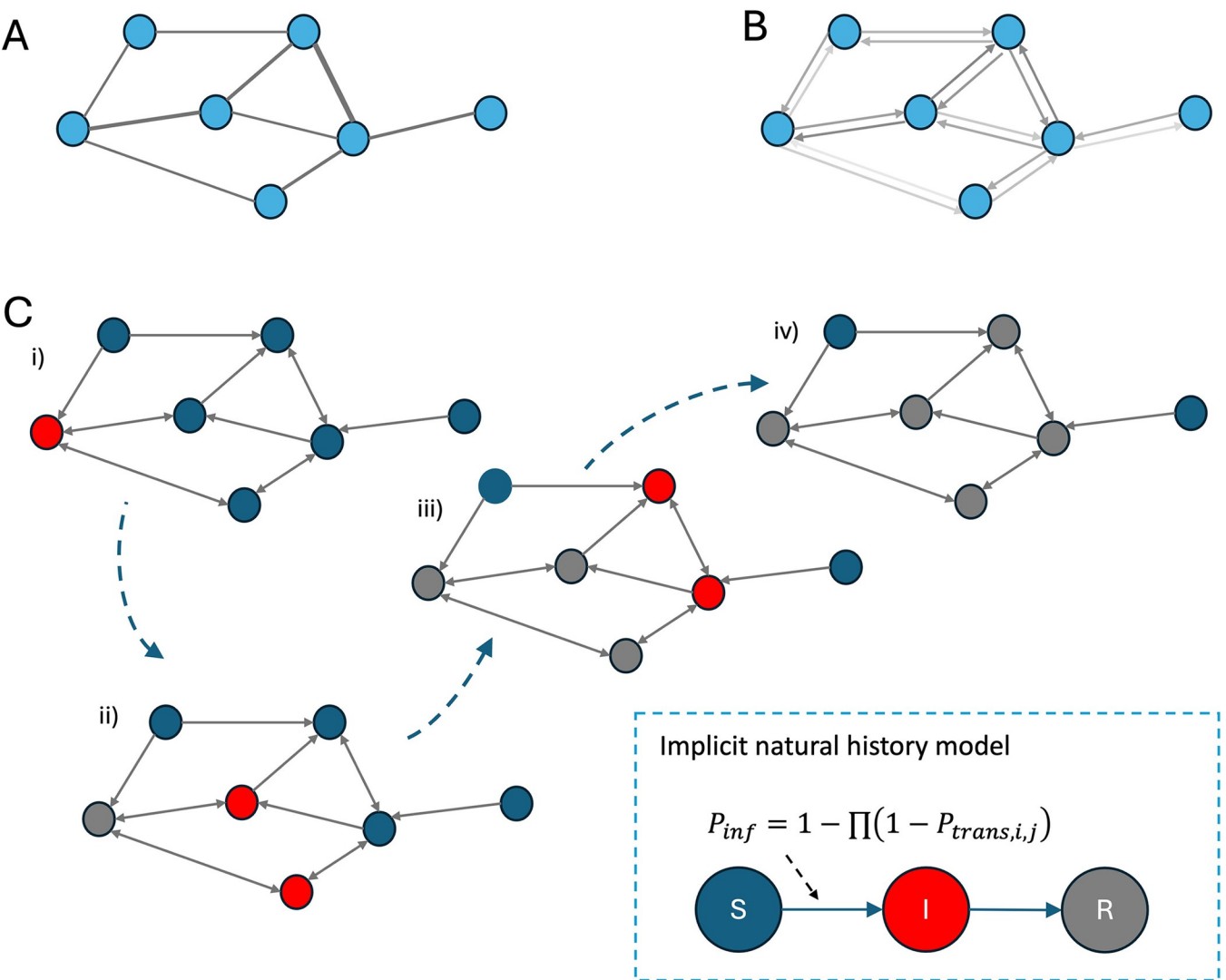

**Fig 2. Schematic of outbreak simulation model.** (A) Shows the contact network between schools (blue circles) where edge weights, indicated by thickness, give the number of unique contact pairs though shared households. (B) Shows the transmission probability network where the directed edge weights give the probability of transmission between schools in each direction, as calculated in Eq 4. (C) Shows binary outbreak networks where directed edges are given a value of 1 or 0, successive generations of an outbreak are shown in panels (i) to (iv), where schools occupy one of 3 epidemiological states (susceptible (S), infected (I), and recovered (R)). In each generation schools connected by an out edge (weight 1) from an infected school are infected in the next generation of the model. This process continues until no outages from infected schools reach susceptible schools.

## Outbreak simulations

In our model, each school can occupy one of 3 states: susceptible, infected, or recovered. Susceptible schools are represented as initially described on the transmission probability network, with immunity profile equal to $1-V_j$. Infected schools are those affected by an outbreak and have a probability of infecting neighbouring susceptible schools ($P_{trans,i,j}$) as defined above. After an outbreak occurred, we assumed that the school had effectively depleted its susceptible population, entering the recovered state where the school could not be re-infected. For each iteration of the model, we sampled a set of vaccine uptake values (per school) from the distributions given by Klinkenberg and colleagues. For each sample, the values of $P_{trans,i,j}$ collectively form a static directed network of transmission probabilities. To create outbreak realisations,

we sampled edges of an equivalent network where edges have a weight of 1 with probability $P_{trans,i,j}$. These binary directed networks represent paths along which transmission can occur in the simulation (Fig 2). For each realisation of an outbreak, one or more schools were set to be in the infected state, outbreaks were then described by trees formed by successive out-edges of the binary network (the out-component).

## Evaluating the model against epidemiological data

To evaluate the ability of the network-based simulations to capture observed measles epidemiology, we compared simulated outbreaks to final estimates of cases from a large outbreak in 2013/14. To do so, the cumulative measles cases per PC4 for the 2013 to 2014 were obtained from the National Registry of Reportable Infectious Diseases (OSIRIS).

Simulations of the outbreak were initiated by placing the 2 schools, which are believed to be the index-schools in the outbreak of 2013/14, in the infected state. All other schools were initiated in the susceptible state. We calculated the total number of students expected to be infected per school by multiplying the final size proportion estimate by the number of students in the school. Then, we allocated infected students proportionally to PC4 areas, based on the proportion of children in each school that live in each PC4 area, which allowed us to compare the simulated outbreaks to the observed historical outbreak. We ran the model 1,000 times and calculated the mean number of cases per PC4 area over all realisations of the simulation. Average model outcomes were compared using receiver operating characteristic (ROC) on PC4-level, based on the presence or absence of cases. To reflect the relative importance of PC4s with higher reported or simulated incidence, we also calculated a weighted ROC (wROC). For this measure, when calculating sensitivity, each PC4 with cases reported is weighted by the proportion of all cases reported in that PC4. Hence, whereas for the unweighted ROC the sensitivity value is the proportion of areas with cases reported that also had cases predicted, for the wROC the sensitivity value is the proportion of cases reported that occurred within PC4s where cases were predicted by the model.

## Evaluating the risk of outbreak posed by each school

We used the transmission model described above to evaluate the relative risk that an outbreak originating in each school poses to the network as a whole. To quantify this risk, we simulated (as described above) 1,000 outbreaks initiated at each school in the network and reported the mean number of schools and children infected.

To establish the relevance of the specific schools independent of their vaccine status, we present the number of schools and children infected by a school by that school's vaccine uptake. To evaluate the sensitivity of these results to the network structure, we compared results from our transmission model with those of alternative model 2 (with a spatially derived network).

All analysis was performed in Python 3 [43]. Network analysis was performed using NetworkX [44]; community detection was performed using the Leidenalg [32] package and evaluation of the partitions was performed using the CDLIB [45] package. All other analyses made use of the scientific python package library [46].

## Results

### Communities in the network

We found that the algorithm converged on a consensus partition after 2 rounds (**Fig K in S1 Text**), this attests to the stability of the initial set of partitions. Indeed, after the first round

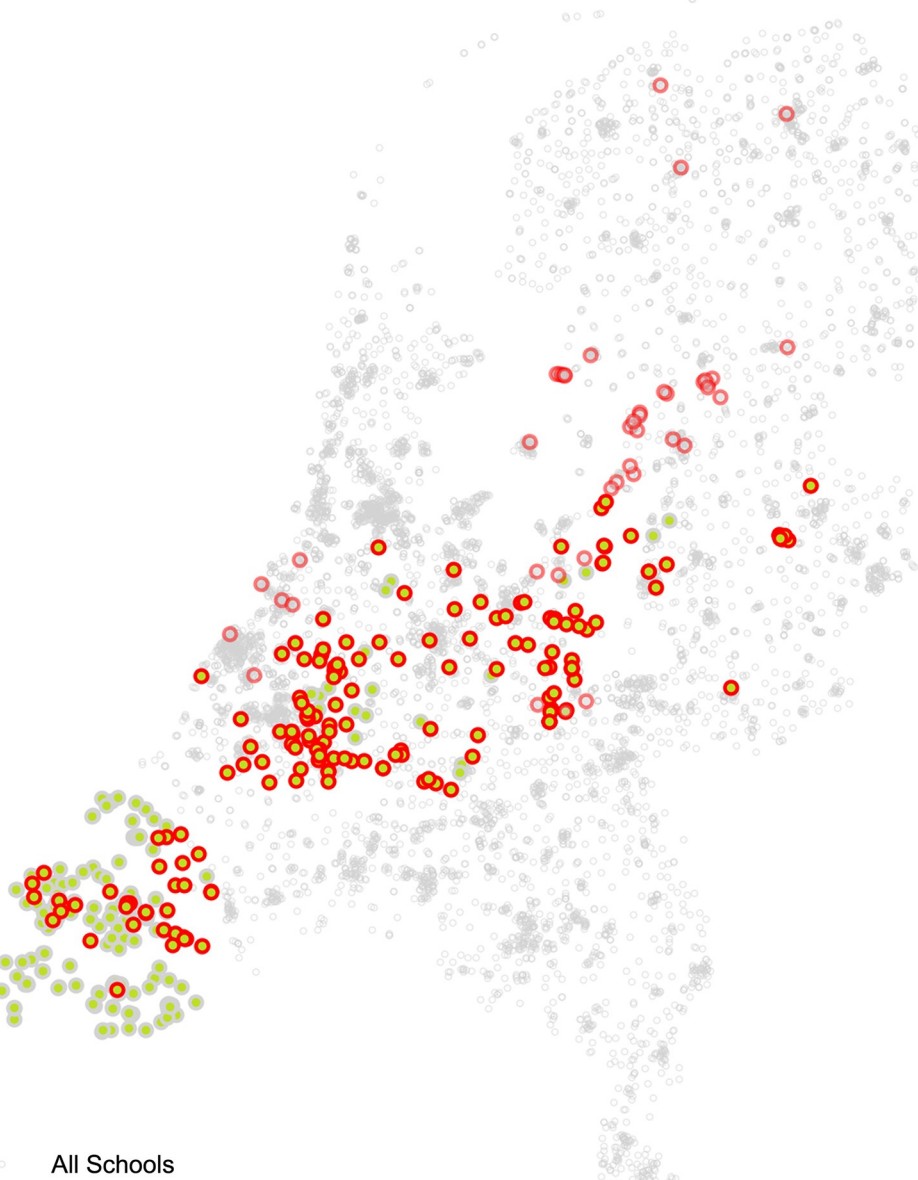

○ All Schools

● Community 9 Not Orthodox Protestant

⊙ Community 9 and Orthodox Protestant

○ Orthodox Protestant not Community 9

**Fig 3. Consensus partition community 9 which is primarily composed of schools in the Zeeland province and schools with the Orthodox Protestant identity (*"Reformatorisch"*) spread across the Nation.**

only 2 unique partitions were found (**Fig I in S1 Text**), which themselves were very similar, with a normalised mutual information score of 0.98 (**Fig J in S1 Text**).

The final consensus partition resulted in mostly geographically organised communities, with high probability of schools in the same province being assigned the same community. In general, any preference of connection between schools of the same identity was not sufficient to overpower the strong geographical component in the communities. The exception was community 9 (Fig 3), which was partially associated with the province of Zeeland (250 schools,

**Table 2. Mean pairwise probability of being partitioned into the same community by school identity and province.**

| Identity | | | | Province | | |
|---|---|---|---|---|---|---|
| Dutch name | English translation | MPP | 95% CI | | MPP | 95% CI |
| Reformatorisch | Orthodox Protestant | 0.55 | (0.546, 0.554) | Groningen | 0.996 | (1.00, 0.996) |
| Overige | Other | 0.333 | (0, 0.776) | Friesland | 0.988 | (0.988, 0.989) |
| Hindoeistisch | Hindu | 0.28 | (0.081, 0.479) | Noord-Holland | 0.950 | (0.949, 0.950) |
| Gereformeerd | Reformed | 0.268 | (0.22, 0.317) | Zeeland | 0.887 | (0.885, 0.889) |
| Interconfessioneel | Interconfessional | 0.233 | (0.196, 0.271) | Limburg | 0.857 | (0.855, 0.858) |
| Gereformeerd vrijgemaakt | Reformed liberated | 0.21 | (0.203, 0.217) | Noord-Brabant | 0.784 | (0.783, 0.785) |
| Evangelisch | Evangelical | 0.149 | (0.105, 0.193) | Drenthe | 0.755 | (0.753, 0.758) |
| Rooms-Katholiek | Roman Catholic | 0.147 | (0.147, 0.147), | Utrecht | 0.724 | (0.722, 0.725) |
| Islamitisch | Islamic | 0.147 | (0.132, 0.162) | Flevoland | 0.684 | (0.682, 0.686) |
| Openbaar | Unaffiliated | 0.129 | (0.128, 0.129) | Overijssel | 0.546 | (0.544, 0.547) |
| Protestants-Christelijk | Protestant | 0.125 | (0.124, 0.125) | Zuid-Holland | 0.442 | (0.441, 0.442) |
| Algemeen bijzonder | General specialised | 0.122 | (0.121, 0.122) | Gelderland | 0.343 | (0.342, 0.344) |
| Antroposofisch | Anthroposophic | 0.118 | (0.11, 0.125) | | | |
| Mean | | 0.28 | | | 0.75 | |

61% of the community). However, an additional 155 schools from other provinces were included in this community including 129 (31% of the community) schools from the Orthodox Protestant ("*Reformatorisch*") identity. In total, 166 (41% of the community) schools associated with the Orthodox Protestant identity were included in this community from 6 different provinces, this represents 79% of all schools of this identity.

The MPP that schools of the same province fell into the same partitioned communities was high with a mean of 0.75 (Table 2). In contrast, the MPP that schools of the same identity were partitioned into the same communities was much lower with a mean of 0.28. There were 2 identities which were excluded from the analysis, the Jewish identity ("*Joods*") and the identity linked to the Moravian Church ("*Evangelische broedergemeenschap*"), both have only a small number of geographically clustered schools in the network resulting in pairwise probabilities of 1.0. Of the remaining, larger identities, the Orthodox Protestant schools had the highest pairwise probability of 0.55. In contrast, Anthroposophic identity schools had a much lower pairwise probability of 0.12 (Table 2).

## Network analysis

On average each school was directly connected to 39.8 other schools through 225.9 contact pairs. These values were much lower for primary schools, 27.5 and 122.1, respectively. Conversely secondary schools had many more immediate neighbours with 109.7 connected schools through 813.9 contact pairs (Fig 4A). Schools with the Orthodox Protestant identity tended to have a higher ratio of contact pairs to the number of schools compared to the rest of the network, whereas schools with the Anthroposophic identity had a more characteristic relationship between the number of neighbouring schools and contact pairs indicating more thinly distributed contact between a larger number of schools.

**Relative connectedness.** For the majority of identities there is a positive homophily index, suggesting that households are more likely to have children in 2 or more schools of the same identity than would be expected at random (Fig 4B).

The 4 identities with the highest CHI were Orthodox Protestant, Anthroposophic, Roman Catholic, and mainstream Protestant, with CHI ranging from 0.62 to 0.12. Notably the 2 school identities with the highest Coleman Index were Orthodox Protestant (Basic Homophily (HI) =

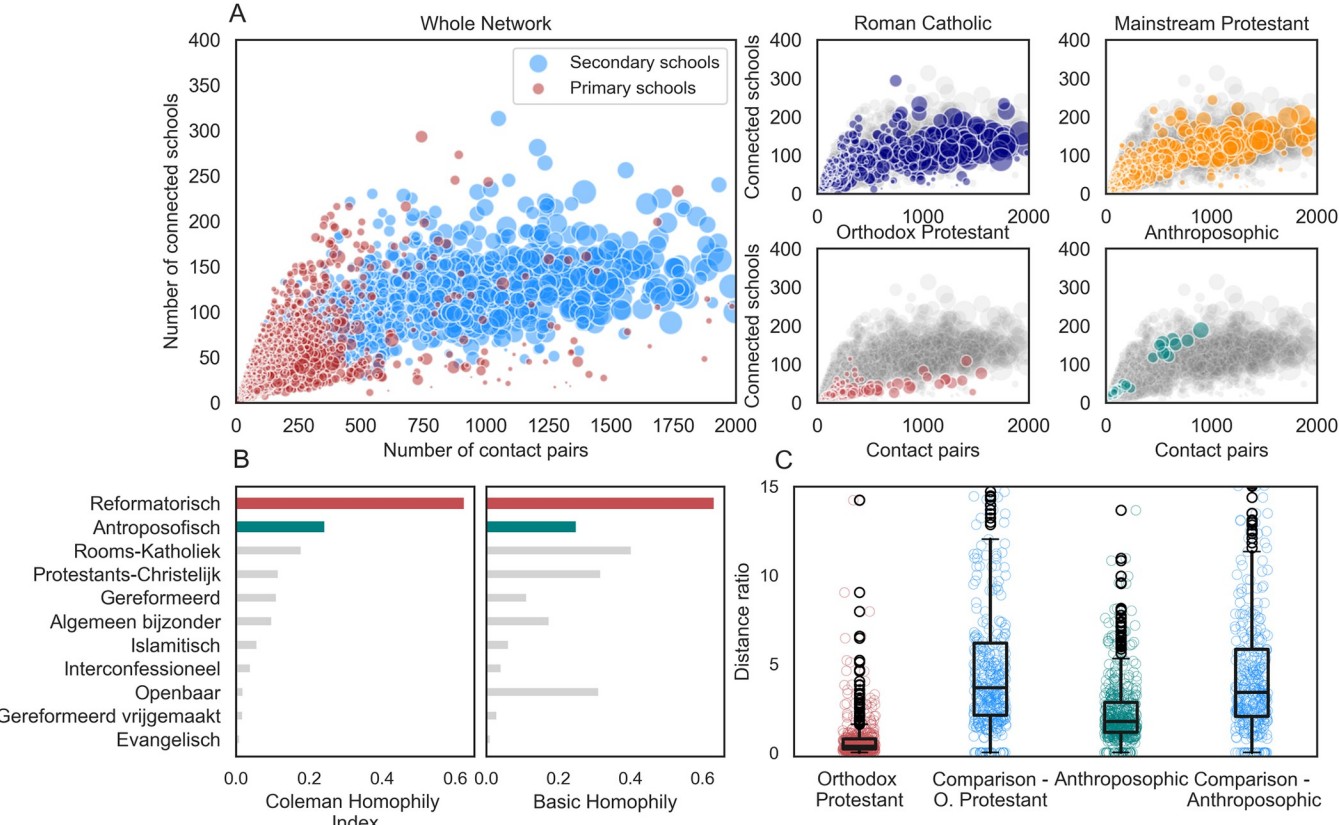

**Fig 4.** (A) The number of contact pairs connected to other schools against the number of connected schools for each school in the network. The main panel shows the primary schools in red and secondary schools in blue. The 4 panels on the right show the 2 largest school identities (protestant and catholic) alongside the Orthodox Protestant ("*Reformatorisch*") and Anthroposophic ("*Antroposofisch*") identities. The rest of the schools in the network are shown in grey for comparison. (B) The 11 identities with the highest CHI. In the left panel, bars show the CHI of each identity. In the right panel, bars show the basic homophily of each identity. Red bars highlight the Orthodox Protestant and Anthroposophic identity schools, where vaccination uptake is known to be low. (C) Boxplot of distance ratio for pairs of Orthodox Protestant and Anthroposophic identity schools and geographically equivalent sample from the rest of the network.

0.63, CHI = 0.62) and Anthroposophic (HI = 0.25, CHI = 0.24), which are the 2 school identities with relatively low vaccine uptake compared to the general Dutch population (Fig 5).

**Distances across the network.** The mean ratio of network to geographic distance was 3.09 pairs$^{-1}$ km$^{-1}$ for the whole network, 0.54 pairs$^{-1}$ km$^{-1}$ for schools with the Orthodox Protestant identity 3.82 pairs$^{-1}$ km$^{-1}$ for schools with the Anthroposophic identity, 3.75 pairs$^{-1}$ km$^{-1}$ for schools with the Roman Catholic identity and 3.17 pairs$^{-1}$ km$^{-1}$ for schools linked to the Protestant identity. This indicates that schools linked to the Orthodox Protestant identity form extended chains of schools linked through households, whereas the other identities are generally as connected as any schools in the whole network.

Moreover, the distance ratio distribution was lower for Orthodox Protestant identity schools (0.54 pairs$^{-1}$ km$^{-1}$ versus 5.18 pairs$^{-1}$ km$^{-1}$) and Anthroposophic identity schools (3.83 pairs$^{-1}$ km$^{-1}$ versus 4.81 pairs$^{-1}$ km$^{-1}$) than their comparison samples (Fig 4). This suggests that in both cases, network paths were shorter between schools with the Orthodox Protestant and Anthroposophic identity than between randomly selected geographically equivalent schools with a different identity.

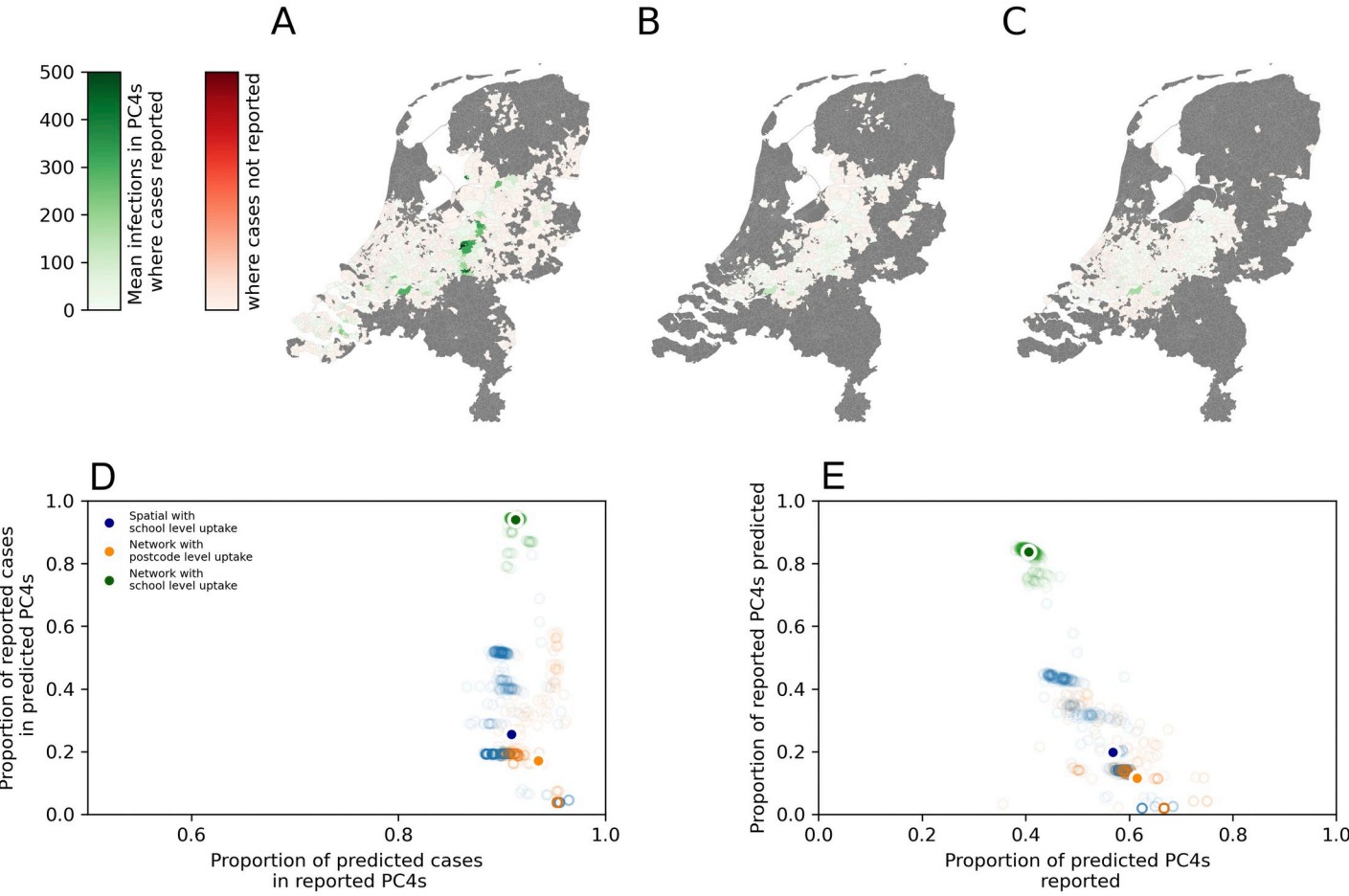

**Fig 5. Mean number of cases across 1,000 simulated in each PC4 region with a reporting rate of 10% (from estimates in literature).** (A) The baseline model: School data network with school level uptake, (B) alternative model 1: School data network with PC4 level uptake, (C) alternative model 2: Spatial network with school level uptake, (D) weighted sensitivity and specificity, and (E) unweighted sensitivity and specificity of the baseline and alternative network models. Geometries of the PC4 areas were provided by GADM (https://gadm.org/about.html).

## Simulation studies

**Comparison to the 2013/14 outbreak.** Using the baseline model (national school data contact network and school level vaccine uptake estimates), 1,000 outbreak simulations with the initial schools set to the 2 schools first identified in the 2013 outbreak resulted in a mean of 24,310 (24,052–24,736 IQR) infections. The geographical distribution of cases was broadly consistent with the reported cases in 2013/14. There was a high likelihood of cases being reported in PC4 areas in the centre of the country and the southwest. There was also a high likelihood of infections in a small region in the north of the country (Fig 5).

When alternative model 1 was used (national school data in combination with the vaccination uptake in schools estimated from PC4 level vaccine uptake), the mean final size of the outbreaks was 624 (25–559 IQR). The cases were distributed in a narrow strip, with a high frequency of cases in the region from the southwest region to the northeast of the central region (Fig 5).

When alternative model 2 was used (the spatially derived contact network and school level vaccine uptake estimates), the final size of the outbreak was 2,708 (2,218–5,087 IQR) cases.

The majority of cases predicted occurred in schools in the central region of the country, with low probability of detecting infection in any other regions (Fig 4C).

Using the unweighted ROC, the mean sensitivity (proportion of PC4s where cases reported that were predicted by the model) was 0.84 (0.76–0.86 95% CI), 0.12 (0.02–0.38 95% CI), and 0.2 (0.02–0.44 95% CI), for the baseline model, alternative model 1 and alternative model 2, respectively (Fig 5D). The mean specificity (proportion of PC4s where cases were predicted that also had cases reported) was 0.4 (0.39–0.42 95% CI), 0.62 (0.45–0.73 95% CI), and 0.57 (0.44–0.67 95% CI) for baseline model, alternative model 1, and alternative model 2, respectively.

Considering the weighted ROC. The mean sensitivity was 0.94 (0.87–0.95 95% CI), 0.17 (0.03–0.53 95% CI), and 0.26 (0.04–0.52 95% CI) for the baseline model, alternative model 1, and alternative model 2, respectively (Fig 4E). The mean specificity was 0.91 (0.91–0.92 95% CI), 0.93 (0.89–0.96 95% CI), and 0.91 (0.88–0.96 95% CI) for the school data network with school vaccination, spatial network with school vaccination, and school data with PC4 level vaccination, respectively.

**Outbreak size by school where the outbreak is initiated.**   The overall risk posed by an outbreak seeded in each particular school was quantified by finding the distribution of final outbreak size. For both the school data and spatial networks, the majority of schools had a very low mean outbreak size as no sustainable transmission was observed in any simulation.

For the full network model, the maximum mean outbreak size was 158 schools and 24,324 children and was seeded in a school with the Orthodox Protestant identity (Fig 6). In general, outbreaks seeded in schools with this identity were larger, particularly where the seed schools had very low vaccination coverage. Outbreaks seeded in schools with an Anthroposophic identity generally remained much smaller, with a maximum of 5 schools and 598 children.

Outbreaks simulated on alternative model 2, with a spatially derived network, tended to be smaller. In comparison to the full network model, there remained a general trend with vaccine uptake; however, in many cases some outbreaks seeded in schools with very low uptake were much smaller. The largest outbreaks using alternative model 2 had a mean of 25 infected schools and 6,645 children infected. In this simulation, the largest outbreaks were seeded in a school with the Orthodox Protestant identity. However, the difference between outbreaks seeded in the schools with the Orthodox Protestant and Anthroposophic identities were much less substantial than for the full network model, with a number of outbreaks seeded in Anthroposophic identity schools becoming larger than that predicted by the model derived from the school data. The largest outbreak seeded in a school with Anthroposophic identity included 23 schools and 5,762 children. Notably outbreaks seeded in certain Anthroposophic identity schools were comparable to those seeded in Orthodox Protestant identity schools with similar vaccine uptake.

## Discussion

We used a national school-household network including primary and secondary schools that represent >99% of school-aged children in the Netherlands as a framework to quantify the outbreak risk of measles given observed vaccine-uptake by school. Doing so identified that large close networks within specific school identities, where schools with the Orthodox Protestant identity connect households (and vice versa) over a large geographical distance in the Netherlands. Our network approach (parameterised on 2013 data) was able to predict accurately (sensitivity of 0.94 and specificity of 0.91) the measles cases per PC4 level as observed in the 2013/14 outbreak in the Netherlands. The total incidence predicted by the model (approx. 24,500 infections) was consistent with estimates from literature (30,000 infections with 77% in

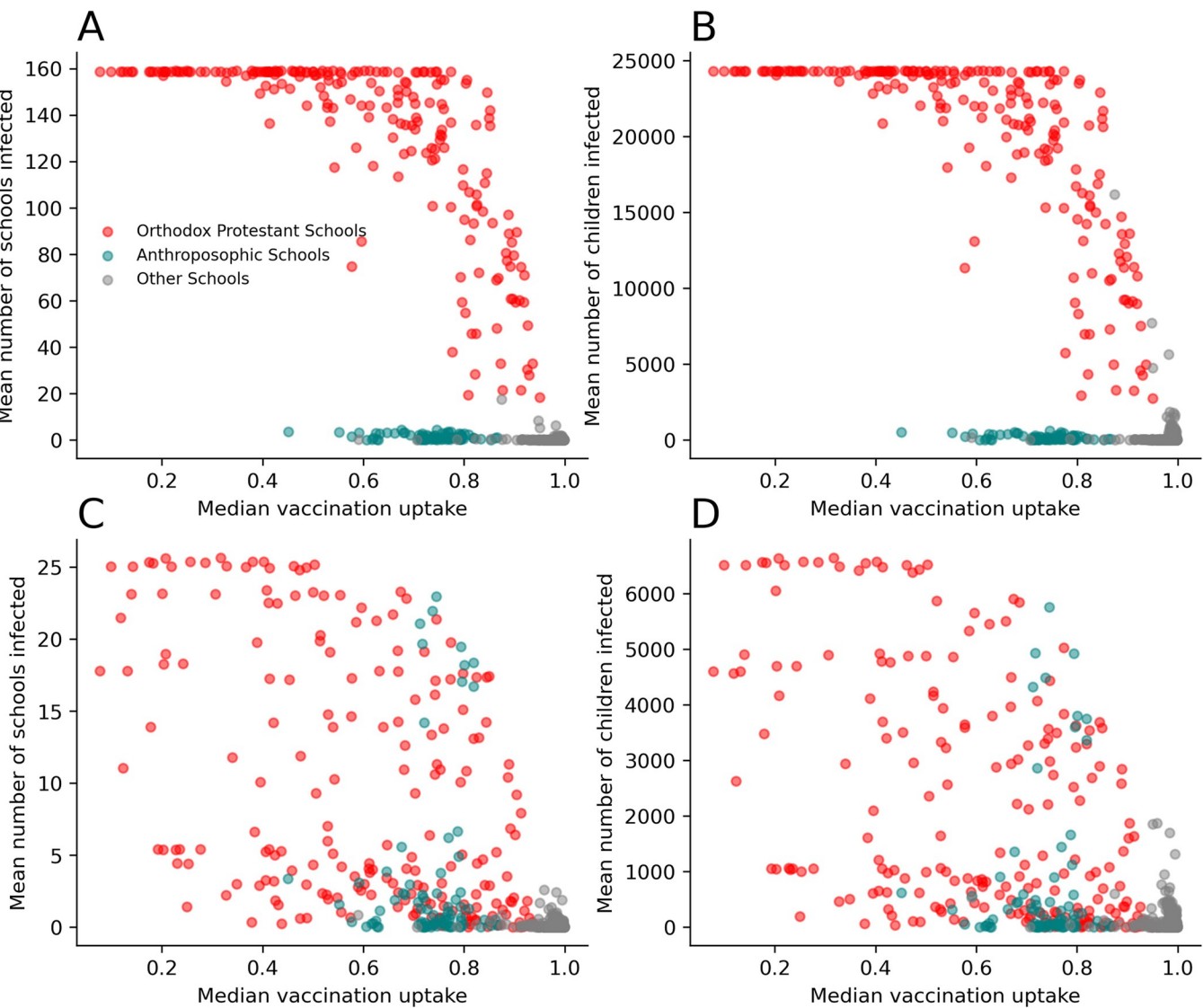

**Fig 6. Mean outbreak final size by school where the outbreak is seeded.** Red points indicate schools with the Orthodox Protestant identity ("*Reformatorisch*"), green points indicate schools with the Anthroposophic identity ("*Antroposofisch*"), grey points indicate other schools. (A) School data-derived network; mean number of schools infected, (B) school data derived network; mean number of children infected, (C) spatially derived model; mean number of schools infected, D) spatially derived model; mean number of children infected.

school-aged children (4 to 17 years old)). This indicates that by incorporating a school-household network on top of school-specific uptake data, we can quantify outbreak risks for measles, the overall outbreak size, the geographical spread of these outbreaks, and identify which particular school contributes to which degree to observed cases per PC4. Our analysis of the network of schools reveals that in particular schools with the Orthodox Protestant identity are much more connected via households than would be expected geographically, evidenced by short network distances when compared to geographical distances. This property is not shared with the Anthroposophic identity schools. The increased connectedness between Orthodox Protestant identity schools, combined with lower MMR coverage, may contribute to the formation of clusters of unvaccinated children, potentially increasing the risk of larger outbreaks of measles, mumps, and rubella among children attending these schools.

Our simulation studies allowed us to quantify the implications of the clustering we identified in the network analysis. Our results have significant implications for understanding the determinants of large outbreaks of measles in the Netherlands.

Firstly, our findings indicate that the distribution of unvaccinated children within particular schools, and the specific links between these schools greatly increase the potential for large outbreaks to occur. With these factors accounted for in the model, outbreaks similar to that observed in 2013/14 can be simulated by accounting for school and household transmission only. This finding suggests that, in a population with immunity provided only by vaccination, outbreaks have a determinable reach, which is not reliant on chance encounters or rare long-range transmission events.

Secondly, our evaluation of the school-network highlights the difference between Orthodox Protestant and Anthroposophic identity schools in how they interact with schools of their own identity compared to the rest of the network. This offers an explanation for the differences in outbreak sizes observed between the Orthodox Protestant and Anthroposophic communities in the past, with outbreaks in Anthroposophic communities typically involving fewer schools, while the 2013 outbreak, which predominantly affected the Orthodox Protestant community, involved a larger number of cases [17,24].

Thirdly, it is evident from the degree distributions of primary and secondary schools that secondary schools are more connected on the network, which is consistent with observations of the school network in England, United Kingdom [29]. This suggests that secondary schools may play a more substantial role in determining the spatial distribution of measles outbreaks than primary schools—a property that could be explored further in future work.

Further, since the variation in outbreak size is due to structural differences in the population, it is likely that future outbreaks in these communities would follow similar patterns, if the structure of the school system remains comparable in years to come. More detailed exploration of these dynamics could be studied by considering the evolution of immunity in schools. For example, our analysis could be repeated on an annual basis utilising the data held in the Dutch vaccination registry and considering the additional immunising effect of previous outbreaks in combination with administrative data of the department of education. Such analysis could assist national and local public health teams in their assessment of risk and subsequently which groups of parents/children to target for national and local campaigns to reduce this risk.

Our approach made some important simplifying assumptions regarding transmission of measles between children: First, the model does not account for possible transmission between children outside school and household (e.g., sport, church activities, or assumes that these contacts can still follow school networks). Further, it is the case that transmission would only occur through other routes (not schools) during school holidays and weekends, these potential transmission routes are not captured in simulations with the model in its current form. These neglected routes of transmission could potentially influence transmission dynamics in a way that this model cannot capture. Additionally, the model does not take into account transmission outside of the school-aged population. Adults and preschool-age infants are likely to contribute to transmission to some degree. In 2013/14, there were 438 cases (19%) in children between 1 and 4, lower than the 819 (30%) and 868 (32%) cases in the 5 to 9 and 10- to 14-year-old age groups, respectively, suggesting less transmission within preschool-age than school-aged children [17]. The presence of preschool institutions in the network would provide additional connectivity on the network which may increase transmission opportunities, particularly between primary schools, however considering the lower contribution of primary schools to connectivity in the network, it might be expected that preschool settings, which tend to be smaller, would provide limited additional transmission opportunity compared to the currently evaluated network.

Secondly, our model does not simulate within-school transmission dynamics, but instead assumes a deterministic final size approximation [39], which occurs with a probability determined by the effective reproduction number in that school. This cannot capture the contribution of outbreaks that do not reach sustainable transmission within schools, but still represent some small risks in terms of infecting other schools with the few pupils that are infected.

Finally, the model works purely on a "generational" basis, with no explicit temporal element. This restricts its use to modelling the overall incidence of an outbreak without modelling the temporal dynamics and therefore preventing comparison of epidemic trajectory between the model and outbreak data.

These limitations, however, do not detract from the findings that the differentiated school system provides a system of contact that can facilitate large outbreaks among unvaccinated children in schools with the Orthodox Protestant identity but not among children in Anthroposophic identity schools.

Further analysis of this network could facilitate the study of other infectious diseases such as mumps and rubella, which are also prevalent among school-aged children within the same socio-religious populations. The model could also be extended to analyse outbreaks of influenza, where a large degree of transmission occurs within school-age children. Another use of this framework could be to evaluate the effectiveness of various other intervention strategies, such as school closure. This method could also be applied in other settings where vaccine uptake is strongly related to particular social groups [14], for example, in the United States, where non-medical vaccine-mandate exemption also has the potential to generate clusters of unvaccinated children [9–12]. However, further application of this framework relies on detailed school data being made available in the relevant settings. Further, our framework could be used to provide parameterisation for more detailed models which explicitly model infection between individual hosts [47–49], or by applying methods to account for within school transmission dynamics using more parsimonious frameworks [50].

## Conclusion

Our results indicate that the tendency for lower MMR uptake in Orthodox Protestant and Anthroposophic communities in the Netherlands may lead to the formation of large clusters of children who are at high risk of measles infection through school and household contact. We found that schools associated with these groups displayed substantial homophily on the school network, indicating higher degree of connectedness than to other schools. By explicitly modelling connections, we can provide valuable insights into the epidemiology of measles in the Netherlands and why it may vary between socio-religious groups. The results of our simulation studies suggest that the network improves our model's ability to describe observed epidemiology from previous outbreaks. High- and long-range network connectedness between schools with the Orthodox Protestant identity, as revealed in our network analysis, may contribute to a higher risk of larger outbreaks in this community compared to schools with the Anthroposophic identity. Between whom connectivity on the network is weaker. This framework could serve as a basis for evaluating risk of large outbreaks of measles in the Netherlands and could be further developed to aid future outbreak response.

## Supporting information

**S1 Text. Supplementary information for "Estimating the risk and spatial spread of measles outbreaks in populations with high MMR uptake: using school-household networks to understand the 2013–2014 epidemic in the Netherlands."**
(DOCX)

## Acknowledgments

The authors wish to thank members of the NIHR HPRU in immunisation and CMMID at LSHTM and the Infectious Disease Modelling Unit at RIVM for feedback. In particular, the authors would like to thank Mark Jit, Petra Klepac, Sebastian Funk, Ada Collis Munday, Leon Danon, and John Read for their insightful discussions.

The views expressed are those of the author(s) and not necessarily those of the NIHR, UK Health Security Agency or the Department of Health and Social Care.

## Author Contributions

**Conceptualization:** James D. Munday, Katherine E. Atkins, Albert Jan van Hoek.

**Data curation:** James D. Munday, Don Klinkenberg, Marc Meurs, Erik Fleur, Albert Jan van Hoek.

**Formal analysis:** James D. Munday.

**Investigation:** James D. Munday.

**Methodology:** James D. Munday, Jacco Wallinga.

**Supervision:** Katherine E. Atkins, Don Klinkenberg, Susan JM Hahné, Jacco Wallinga, Albert Jan van Hoek.

**Visualization:** James D. Munday.

**Writing – original draft:** James D. Munday.

**Writing – review & editing:** James D. Munday, Katherine E. Atkins, Don Klinkenberg, Marc Meurs, Erik Fleur, Susan JM Hahné, Jacco Wallinga, Albert Jan van Hoek.

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
