## [Editor Report · Decision Letter 0]

21 Feb 2024

Dear Dr Munday, 

Thank you for submitting your manuscript entitled "Anticipating the risk and spatial spread of measles in populations with high MMR uptake: Using school-household networks to understand the 2013 - 2014 outbreak in the Netherlands." for consideration by PLOS Medicine.

Your manuscript has now been evaluated by the PLOS Medicine editorial staff as well as by an academic editor with relevant expertise and I am writing to let you know that we would like to send your submission out for external peer review.

I also have a small editorial request. Upon our initial read we note that your write ""correlations between religious faith and vaccination refusal." which is rather a broad statement. I like to suggest you rephrase this sentence, something along the lines that MRR vaccine uptake tend to be lower in *add specific school groups here*. 

Please re-submit your manuscript within two working days, i.e. by Feb 23 2024 11:59PM.

Feel free to email me at kjanin@plos.org if you have any queries relating to your submission.

Kind regards,

Katrien G. Janin, PhD

Senior Editor

PLOS Medicine

---

## [Decision Letter · Decision Letter 1]

12 Apr 2024

Dear Dr. Munday,

Thank you very much for submitting your manuscript "Anticipating the risk and spatial spread of measles in populations with high MMR uptake: Using school-household networks to understand the 2013 - 2014 outbreak in the Netherlands." (PMEDICINE-D-24-00549R1) for consideration at PLOS Medicine. 

As you will see, the reviewers were positive about the paper but, they raised a number of questions about specific study details and the methodological approach. After discussing the paper with the editorial team and an academic editor with relevant expertise, I’m pleased to invite you to revise the paper in response to the reviewers’ comments. We plan to send the revised paper to some of all of the original reviewers*, and of course we cannot provide any guarantees at this stage regarding publication. 

[LINK]

When you upload your revision, please include a point-by-point response that addresses all of the reviewer and editorial points, indicating the changes made in the manuscript and either an excerpt of the revised text or the location (eg: page and line number) where each change can be found. Please submit a clean version of the paper as the main article file and a version with changes marked should as a marked-up manuscript. Please also check the guidelines for revised papers at http://journals.plos.org/plosmedicine/s/revising-your-manuscript for any that apply to your paper. 

We ask that you submit your revision by the 3rd of May, 2024. However, if this deadline is not feasible, please contact me by email (kjanin@plos.org), and we can discuss a suitable alternative.

Please don’t hesitate to contact me directly with any questions (kjanin@plos.org). If you reply directly to this message, please be sure to ‘Reply All’ so your message comes directly to my inbox

We look forward to receiving your revised manuscript. 

Sincerely,

Katrien Janin, PhD

PLOS Medicine

plosmedicine.org

Comment from the Academic Editor

The methods will need to be made more accessible. (Please note that we require methods to be describe in sufficient detail for another researcher to replicate the study)

Wasn't clear to if the models allowed for mixing between primary and secondary schools, e.g. households with siblings in primary and secondary schools (I would assume so, but could be more clearly stated).

ROC estimates for sensitivity and specificity from models need to report uncertainty.

In the Discussion it is also important that the authors reflect appropriately on generalisability outside the Netherlands. (mention of England only, but should consider how this methodology could be applied in a wider range of settings, and what adaptions might be needed).

Overall, it is important to ensure that the authors are mindful to not blame specific denominations for "causing" outbreaks, despite their low vaccine uptake and importance to transmission identified in models. This can be done by being careful in wording (e.g. "... schools seeded outbreaks..." etc

Figure 2 will need a bit of work to improve legibility and interpretation

Comment from the editorial team:

The manuscript sparked a lively debate in terms of how the study is currently presented, particularly with regard to the focus on religious affiliation of the included schools. The consensus among the editorial team was that there is potentially too much emphasis placed on the religious affiliations and that this might detract/distract from the main point of the study regarding the impact of unvaccinated children (in a more general sense) on outbreaks. 

We were also concerned that the presentation might have the potential to offend readers by suggesting that people who subscribe to a certain religion are more likely to be vaccine hesitant (eg, line 328: “…Orthodox Protestant (Reformatorisch) and Anthroposophic (Antroposofisch) … the two faith identities with populations who systematically refuse vaccination.”

Please feel free to contact me directly at kjanin@plos.org if you like to discuss the above further.

In line with the Academic editor, we also feel it important to review the languages used around specific denominations and be mindful not make overgeneralized statements, as noted above. 

We also felt it would be helpful if you could provide more context of the general Dutch school system (in the Introduction) for the benefit of our international readership.

For the model, we feel that the temporal dynamics of the outbreak are important (per the comments of Reviewer #1) and need to be carefully considered in your revision

Table 1: Dutch labels are used for Denomination descriptions. Please provide descriptors in English for the benefit of our international readership (feel free to also keep the Dutch labels) 

Please capitalise 'Orthodox' when describing 'Orthodox Protestant" religion (as part of the proper name). Please check and amend.

TITLE:

Please substitute the word ‘Anticipating’ with ‘Estimating' or something similar (given this is a modelling study).

ABSTRACT:

Please structure your abstract using the PLOS Medicine headings: Background, Methods and Findings, Conclusions. Please remove all other subheaders.

Abstract Background:

Provide the context of why the study is important. The final sentence should clearly state the study question.

Abstract Methods and Findings:

Please include the study design

In the last sentence of the Abstract Methods and Findings section, please describe the main limitation(s) of the study's methodology.

AUTHORS SUMMARY

Ideally each sub-heading should contain 2-3 single sentence, concise bullet points containing the most salient points from your study.

In the final bullet point of ‘What Do These Findings Mean?’ Please include the main limitations of the study in non-technical language.

ACKNOWLEDGMENTS/ DECLARATIONS

Please remove all statements apart from acknowledgements, author contributions and abbreviations from the end of the main manuscript and include these only in the relevant parts of the manuscript submission form. Funding, competing interest, and data availability will be compiled as metadata.

STUDY DESIGN: 

Of all authors who submit a modelling study we ask for inclusion of specific items, derived from Geoffrey P Garnett, Simon Cousens, Timothy B Hallett, Richard Steketee, Neff Walker. Mathematical models in the evaluation of health programmes. (2011) Lancet DOI:10.1016/S0140-6736(10)61505-X.

Please ensure all the items listed below are included with your manuscript. Please review the list below and confirm/revise as necessary:

i) Please provide a diagram that shows the model structure, including how the disease natural history is represented, the process and determinants of disease acquisition, and how the putative intervention could affect the system.

ii) Please provide a complete list of model parameters, including clear and precise descriptions of each parameter, together with the values or ranges for each, with justification or the primary source cited, and important caveats about the use of these values noted.

iii) Please provide a clear statement about how the model was fitted to the data [including goodness-of-fit measure, the numerical algorithm used, which parameter varied, constraints imposed on parameter values, and starting conditions].

iv) For uncertainty analyses, please state the sources of uncertainties quantified and not quantified [can include parameter, data, and model structure].

v) Please provide sensitivity analyses to identify which parameter values are most important in the model. Uncertainty estimates seek to derive a range of credible results on the basis of an exploration of the range of reasonable parameter values. The choice of method should be presented and justified.

vi) Please discuss the scientific rationale for this choice of model structure and identify points where this choice could influence conclusions drawn. Please also describe the strength of the scientific basis underlying the key model assumptions.

DATA AVAILABILIY:

We note your write: “The measles case data is available on request: datastewards@rivm.nl. … Measles case data is not publicly available but is available on request from RIVM. Requests should be made to

osiris.aiz@rivm.nl.” Please clarify where researcher can gain access to the measles case data. 

For data that are not freely available, please describe briefly the ethical, legal, or contractual restriction that prevents you from sharing it.

As a guide: PLOS Medicine requires that the de-identified data underlying the specific results in a published article be made available, without restrictions on access, in a public repository or as Supporting Information at the time of article publication, provided it is legal and ethical to do so. 

"The Data Availability Statement (DAS) requires revision. For each data source used in your study: 

Comments from the reviewers:

Reviewer #1: Thank you for the invitation to review the manuscript by Munday et al. Given school-specific MMR vaccine uptake, the authors created a national school-household network to quantify the outbreak risk of measles in the Netherlands. By simulating measles outbreaks on the network parameterized on 2013 data, the authors were able to explain the observed size and spatial distribution of 2013/14 outbreak, and identified that the highly connected orthodox Protestant schools associated with low vaccine uptake contribute to large outbreaks. The findings are important for understanding why measles outbreaks still occur in countries with high MMR vaccine uptake, and for archiving the WHO goals of eliminating measles.

However, I am concerned about whether it is adequate for reproducing the historical outbreak by comparing only the final size and spatial distribution without considering the epidemic curve. The authors acknowledge in the limitations that the current model cannot model temporal dynamics, but why can the results have important implications for understanding the timing of large measles outbreaks in the Netherlands (Lines 421-422)?

The authors think that their analysis can be repeated on an annual basis to assess the outbreak risk (Line 410). However, in the following discussion on implications, the authors also believe that the future outbreaks are likely to follow similar patterns among communities, if the population structure remains comparable in years to come (Lines 442-444). I am a bit confused whether it is necessary for the public health authorities to conduct the analysis annually.

Moreover, the manuscript is full of formatting typos, including capitalization, punctuation, abbreviations, references etc., which the authors should carefully proofread. Below are those typos to be fixed.

Line 37, change "Network" to "network".

Lines 51-52, add the following reference to support the statement that many countries are facing challenges in Measles outbreaks.

https://www.nature.com/articles/d41586-024-00265-8

Lines 66-67, the sentence "each of these… were in children" is difficult to understand. Please improve it.

Line 112, change "school aged" to "school-aged".

Line 139, change "Supplementary" to "supplementary".

Line 147, there are two "the".

Line 160, the parentheses for equations (1) and (2) are incomplete. Add a period "." After "denomination".

Lines 166-168, variable i should be italic.

Line 170, the citation "Donker et al" misses "." and the year.

Line 175, no space between figure number and panel label.

Line 176, "I" should be lowercase and italic i; This also applies to "i + 1".

Lines 192-193, change "supplementary information" to "supplementary material".

Line 193, the citation "Munday et al." misses the year.

Line 195, change "Susceptible" to "susceptible", "Infected" to "infected".

Line 197, the citation "Klinkenberg et al" misses "." and the year.

Line 206, in equation (4) there is a variable q without definition.

Line 216, no space between figure number and panel label.

Line 231, change "Details" to "details".

Line 237, V should be Vj.

Line 242, "et. al" should be "et al.".

Lines 254 and 503, "disease" should use the plural form "diseases".

Line 272, add "that" after "risk".

Lines 274, 350 and 694, use a comma delimiter for 1000. This also applies to 1241 and 1619 (Line 361).

Line 279, change "Alternative Model" to "alternative model".

Line 297, there are two "of".

Line 310-311, for Table 1 title, change "School" to "school", "Region" to "Province".

For Table 1 content, change "MEAN" to "Mean". 

Although the expanded form of MPP is provided in the List of abbreviations (Line 502), please provide the expansions of MPP and 95% CI in the main text. This also applies to the following CHI (Line 326), BH (Line 329) and IQR (Line 351).

Line 317, change "figure 4 A" to "Figure 4A", "orthodox" to "Orthodox".

Line 324, delete "than expected" since there is "than would be expected at random" at the end of this sentence.

Line 325, change "figure 4 B" to "Figure 4B".

Line 327, change "Mainstream" to "mainstream", "With" to "with".

Line 336, change "The" to "the".

Line 339, I cannot find the "scatter plots of network distance against geographic distance". Please insert a reference to the figure.

Lines 345-347, how 5.18×10-3 and 4.81×10-3 are compared with 0.54 pairs-1km-1 and 3.83 pairs-1km-1 as they have different units and magnitudes?

Line 349, change "National" to "national".

Line 356, change "Alternative" to "alternative", and delete ",".

Line 360, change "Alternative" to "alternative".

Line 363, change "figure" to "Figure".

Lines 371-372, change "School" to "school", "Spatial" to "spatial".

Line 379, change "a" to "an".

Line 393, change "some" to "certain".

Lines 399 and 456, change "school aged" to "school-aged".

Lines 406-407, move "(4-17 years old )" to the first occurrence of "school-aged children" in Line 112.

Line 408, delete "-" from "outbreak-risks".

Lines 431, 433, 434 and 485, change "anthroposophic" to "Anthroposophic".

Line 438, change "this" to "which", and add reference to support the "observations of the school network in England, UK".

Line 455, move "the" before "5-9". 

Line 456, change "pre-school age" to "preschool-age".

Lines 457 and 460, change "pre-school" to "preschool".

Line 459, change "since" to "considering".

Line 460, move "tend to be smaller" between "settings" and ",".

Line 466, "risk" should use the plural form "risks".

Line 479, change "school age" to "school-aged".

Lines 482 and 513, change "this" to "which".

Line 501, MMR is only the abbreviation of Measles, Mumps and Rubella.

Line 506, change "Weighted" to "weighted".

Line 514, change "ministry" to "Ministry".

Line 539-541, change "AJVH" to "AJvH".

Line 539, delete "," before "conceived".

Line 671, change "," to ".".

Line 677, add "[Here labelled "Reformed"]" after "(Reformatorisch)".

Increase the font size of texts in Figure 2 and Figure 4.

Line 682, change "," to ".", "the" to "The".

Line 682, add "and" before "the rest".

Line 685, change "in" to "In"; add "," after "panel".

In the left panel of Figure 4B, change the x-axis label "Coleman Index" to "Coleman Homophily Index".

Line 688-689, delete "[Here labelled "Reformed"]".

Line 695, change "the" to "The"; delete "." after "uptake".

Line 697, add "," after "uptake" and after "specificity".

Reference 8 has been published. Please update.

References 14 and 20 are duplicated.

References 19 and 21 are duplicated.

Supplementary Information

Page 1, change "leiden" to "Leiden".

Variable C should be italic, C.

Add "as" after "expressed".

Page 2, variable i should be italic, i.

Line 2, k_iC^in. should be k_iC^out; 8 should be i; P should be C.

Line 5, change "Is" to "This is".

Page 3, caption of Figure S1, change "+" to γ; add "." after "respectively". 

Page 4, Line 1, the "school j" should be "school i".

Line 7, change "we" to "We".

Line 8, change the reference number 19 to 34 in main text. Also, double check the reference number 19 in the last line.

Line 11, variable i should be italic, i.

Line 16, i.e., the last line, change "of" to "for". 

In main text, a gaussian offspring distribution is assumed; however, here a Poisson distribution is assumed. Which one is correct?

Page 5, Line 1, variable i should be italic, i.

Line 9, change "A" to "a".

In Table S1, change "Denomination" to "denomination", "Mainstream protestant" to "Mainstream Protestant".

Page 7, caption of Figure S2, change "black" to "Black".

Page 8, Line 7, add space between "and" and "10".

Page 9, the texts in Figure S4 are not clear. Please increase the resolution of this figure.

For the figure caption, change "estimates" to "Estimates", "Orthodox" to "orthodox"; the citation "Klinkenberg et al." misses the year.

Reviewer #2: Please see comments in attachment.

Reviewer #3: 

This is an interesting paper concerned with an important topic and the network analysis is useful on its own and well linked to the transmissibility element of this problem. I have some comments that require clarification and amendment.

Please add the evidence for the q=0.5 assumption. This seems like a crucial component, driving the transmission in the overlapping part.

The Poisson distribution for the within school contact rate (supplement end of page 4) is presumably a Poisson process like in typical epidemic models?

It would be useful to summarise in a table the parameters (for each of the 3 models) that are actually calibrated to fit the observed outbreak data.

If feasible this table may also include the network key characteristics, like the λ parameter mentioned in supplement page 7 (above fig S2).

Is the Ro=15 value assumed fixed with no uncertainty?

The Gaussian distribution is somewhat unusual for the offspring, have the authors tried distributions defined on the positive (and perhaps integer) numbers?

The authors do not refer to the large methodological/statistical/theoretical literature and the connections of their model to related models like Britton, Kypraios T. and O'Neill P.D. (2011), Statistical models for epidemic models with three levels of mixing and the series of papers on Network epidemic models with two levels of mixing by Frank Ball and Peter Neal

Which of these (very detailed) data are publicly available?

It would be useful to clarify all the aspects (code+data) of the reproducibility of this work.

1. Please upload any figures associated with your paper as individual TIF or EPS files with 300dpi resolution at resubmission; please read our figure guidelines for more information on our requirements: http://journals.plos.org/plosmedicine/s/figures. While revising your submission, please upload your figure files to the PACE digital diagnostic tool, https://pacev2.apexcovantage.com/. PACE helps ensure that figures meet PLOS requirements. To use PACE, you must first register as a user. Then, login and navigate to the UPLOAD tab, where you will find detailed instructions on how to use the tool. If you encounter any issues or have any questions when using PACE, please email us at PLOSMedicine@plos.org. 

To submit your revised manuscript please use the following link: 

[LINK]

---

## [Decision Letter · Decision Letter 2]

20 Aug 2024

Dear Dr. Munday,

Thank you very much for re-submitting your manuscript "Estimating the risk and spatial spread of measles in populations with high MMR uptake: Using school-household networks to understand the 2013 - 2014 outbreak in the Netherlands." (PMEDICINE-D-24-00549R2) for review by PLOS Medicine.

I have discussed the paper with my colleagues and the academic editor and it was also seen again by the reviewers. I am pleased to say that provided the remaining editorial and production issues are dealt with we are planning to accept the paper for publication in the journal.

[LINK]

We look forward to receiving the revised manuscript by Aug 27 2024 11:59PM.   

Sincerely,

Katrien Janin, PhD

Senior Editor 

PLOS Medicine

plosmedicine.org

Requests from Editors:

Thank you for your detailed response to the editors' and reviewers' comments. I have discussed the paper with my colleagues and the academic editor, and it has also been seen again by the original reviewers. The changes made to the paper were satisfactory to the reviewers, the academic editor and the editorial team. We find that the manuscript has much improved, thank you!

Only a few minor issues remain

1. MAIN COMMENT: The editorial team concurs with reviewer 1, that some copy edit issues remain. Please have thorough copy edit of your manuscript. In places, 'emotive language' remains and we invite you to reduce it as much as possible to data driven and factual statements (see e.g. point 3). 

2. At line 89-90, "This rise is driven by parental hesitancy towards vaccines and reduced MMR vaccination ... " I would rephrase to say “driven in part by” (otherwise too much of an absolute statement). Perhaps also include guardians but leave that op to you. If you can veer away from the word 'hesitancy' that would also be welcomed. 

3. Line 91-92, "This increase in unprotected children raises fears of increased measles outbreak risk as has been documented in the past for measles and other highly infectious pathogens" - we suggest: The increase in unvaccinated children increases the risk of measles outbreak risk as has ..." 

4. Line 98-99 "Apart from hindering efforts to eliminate measles ... " please rephrase and use alternative phrasing for 'hindering'. In a way, I am not sure it is needed and find that "Measles outbreaks pose immediate risks to vulnerable populations, including young children who can't yet receive vaccinations and people with certain medical conditions." works quite well without the 'Apart from hindering efforts to eliminate measles ' part. I leave it up to your discretion. 

5. Line 102 - 103, "The phenomenon of clustering of unvaccinated individuals within social groups is particularly clear in the Netherlands, where the ...". We like to suggest: ""The phenomenon of clustering of unvaccinated individuals within social groups is well documented in the Netherlands [add references here], where the ... [also needs a reference for the number you have included].

6. Line 109, philosophies[16,17]. A space seems to be missing between 'philosophies' and '[16,17]'.

7. This is a kind reminder that Supplementary materials 

Please note that supplementary materials are not checked and will be posted as supplied by the authors. Therefore, please double check. Please cite your Supporting Information as outlined here: https://journals.plos.org/plosmedicine/s/supporting-information - Please note you may use almost any description as the item name of your supporting information as long as it contains an "S" and number. For example, “S1 Appendix” and “S2 Appendix,” “S1 Table” and “S2 Table. Please ensure each supplementary material has a call out (link) from your main manuscript. 

Comments from Reviewers:

Reviewer #1: The authors have substantially addressed my concerns and fixed the typos as suggested. The manuscript has been improved. 

Please refer to the attached manuscript for my additional edits of typos.

Reviewer #2: I thank the authors for their careful consideration of all reviewers' comments and feel satisfied with their changes and clarifications. There are a few typos still floating about, but nothing that would significantly subtract from the paper's message.

Reviewer #3: The authors have revised their paper and this now represents an improved manuscript, acceptable for publication

[LINK]

---

## [Editor Report · Decision Letter 3]

27 Aug 2024

Dear Dr Munday, 

On behalf of my colleagues and the Academic Editor, I am pleased to inform you that we have agreed to publish your manuscript "Estimating the risk and spatial spread of measles in populations with high MMR uptake: Using school-household networks to understand the 2013 - 2014 outbreak in the Netherlands." (PMEDICINE-D-24-00549R3) in PLOS Medicine.

*I do have one small-ish final request: Where you write "This increase in unprotected children increases

measles outbreak risk as has been documented ... to consider rewording to "This increase in unprotected children increases the risk of measles outbreaks as has been documented ...

PRESS

Sincerely, 

Katrien G. Janin, PhD 

Senior Editor 

PLOS Medicine